# Detection of Circulating Tumor Cell-Related Markers in Gynecologic Cancer Using Microfluidic Devices: A Pilot Study

**DOI:** 10.3390/ijms24032300

**Published:** 2023-01-24

**Authors:** Kim-Seng Law, Chung-Er Huang, Sheng-Wen Chen

**Affiliations:** 1Department of Post-Baccalaureate Medicine, National Chung Hsing University, 145 Xingda Road, Taichung City 40277, Taiwan; 2Department of Obstetrics Gynecology, Tung’s Taichung Metroharbor Hospital, 699 Taiwan Boulevard, Section 8, Wuchi, Taichung City 43522, Taiwan; 3CytoAurora Biotechologies Inc., Hsinchu Science Park, Hsinchu 30261, Taiwan; 4Department of Electrical Engineering, National Chung Cheng University, Chiayi 621301, Taiwan

**Keywords:** BioChips, breast cancer, circulating tumor cells, cervical cancer, ovarian cancer, endometrial cancer, microfluidic devices

## Abstract

The detection of circulating tumor cells (CTCs) is an emerging strategy for the early detection, prognostication, and identification of recurrent cancer. The clinical utility of CTC detection has been established, but few studies have employed this strategy for the detection of gynecologic cancers. Here, we present a novel, biochip-based microfluidic device for the detection of CTCs in gynecologic cancers. The study cohort included three patients with cervical cancer, eight with endometrial cancer, two with ovarian cancer, two with breast cancer, and one with vaginal small cell carcinoma. Four cancer type-specific molecular markers (PanCK, GATA3, HER2, and HE4), as well as CD13, were used for prognostication and recurrence detection, along with downstream genomic analysis. GATA3 and HER2 were markedly expressed in the patients with cervical cancer, and this expression was strongly correlated with the early detection of recurrent disease. All four molecular markers were expressed preoperatively in the patients with endometrial cancer, and the re-expression of different markers was observed at follow-up before recurrence was confirmed. CD13 was identified as an alternative prognostic marker for both cervical and endometrial cancer. Our pilot study indicated that the novel CTC detection system can be used for prognostication and early detection of disease recurrence, which needed further investigation.

## 1. Introduction

Cancer is the most prevalent cause of global mortality, with the majority of deaths attributed to metastatic disease [1]. A survey has shown that malignant tumors will be the major cause of death worldwide by 2030, expected to grow to 20.3 million new cancer cases and 13.2 million deaths [2]. Early detection and identification of early recurrence are essential for preventing lethal consequences; however, effective screening tools are not available for every cancer type.

Circulating tumor cells (CTCs) are cells from the original tumor that enter the bloodstream, where they interact with the microenvironment and ultimately extravasate, resulting in metastasis. The detection of CTCs might serve as a tool for improving disease management by enabling the early detection of cancer, prediction of treatment outcomes, and early detection of recurrence [3,4,5,6,7].

The detection of CTCs relies mainly on the detection of epithelial surface markers, namely EpCAM. Currently, CELL SEARCH is the only FDA-approved kit on the market. Several studies have reported the clinical utility of CTC detection, with initial success observed in breast, colorectal, and prostate cancer. However, few articles have discussed the utility of CTC detection in gynecologic cancers [8,9,10,11,12,13,14,15].

The current CTC detection technologies are mainly divided into magnetic beads, microfluidic, and size-based. The magnetic bead method is based on the Cellsearch platform as the mainstream of research, but studies have questioned its low capture rate of CTCs. The method of cell size screening carries the risk that small size CTCs cannot be captured. Microfluidic technology has the advantages of high capture rate and easy operation, and it can locate the cell position with AI image recognition software, which can pick up CTCs accurately and provide high-purity target cell biological information for subsequent molecular analysis.

Advances in microfluidic device technology have enabled the detection of rare CTCs more efficently and the identification of several cancer type-specific markers; additionally, epithelial–mesenchymal transition surface markers provide an in-depth evaluation of cancer. Here, we describe the application of a novel method for the detection of CTCs in gynecologic cancer cases and demonstrate its utility in clinical management.

## 2. Results

A total of seventeen patients, including three with breast cancer, three with cervical cancer, eight with endometrial cancer, two with ovarian cancer, and one with vaginal cancer, were included in our study. Patient demographic characteristics are listed in Table 1 and Table 2. Lymph node status and histologic grading, as well as histology, are shown in Table 3 and Table 4. Table 5 shows the total CTC count in different cancers, along with the expression of CD13.

The three cervical cancer patients included one stage IIIC1 endocervical adenocarcinoma patient (P4), one stage IIIC1 squamous cell carcinoma patient (P18), and one stage IIB squamous cell carcinoma patient (P19). These patients exhibited significant expression of HER2 markers, compared to healthy cohorts (Figure 1). The marked expression of HE4 and HER2 was noted in P4, who underwent pretreatment for stage IIIC1 endocervical adenocarcinoma. The expression of both markers was normalized by the first follow-up (Figure 2). Re-expression of HER2 was noted at the fourth follow-up, with negative clinical findings. Normal tumor marker expression was observed, and a PET scan of the lungs produced equivocal findings (score 2). Twelve months after the initial diagnosis, a metastatic lesion was observed after the patient underwent video-assisted thoracoscopic surgery, which yielded histologic evidence of adenocarcinoma metastasized from the cervix. The expression of the CD13 marker was increased at the first follow-up, peaked at the second follow-up, and persisted through the third follow-up (Figure 2) Table 1 shows patient demographic characteristics. Patients were numbered according to the date of enrollment.

Patient 18 (P18) was initially diagnosed with stage IIIC1 squamous cell carcinoma, with persistent elevated serum SCC after CCRT. She underwent adjuvant hysterectomy and bilateral salpingo-oophorectomy, and no histological evidence of disease recurrence was observed. The patient was noted to have increased expression of CD13 at the first follow-up and mild expression of GATA3/PanCK at the second follow-up (Figure 2). A PET scan revealed avid uptake in the neck lymph node, and lymph node dissection confirmed the metastasis of cervical cancer.

P19, diagnosed with stage IIB squamous cell carcinoma, exhibited initial expression of GATA3, PanCK, and HER2. She underwent CCRT, with no evidence of disease at the time of follow-up. However, all three markers, which exhibited marked expression at the first follow-up, indicated no evidence of disease at the time of writing. Marked expression of CD13 was noted, which was normalized by the first follow-up and increased again by the second follow-up (Figure 2).

The eight endometrial cancer patients included three stage IA patients (P3, P5, and P21), one stage IB patient (P14), one stage IIIA patient (P10), one stage C1 patient (P13), one stage IIIC2 patient (P11), and one stage IVB patient (P7). All patients exhibited endometrioid histology, except for the carcinosarcoma patient (P13), and all exhibited preoperative overexpression of PanCK, GATA3, HE4, and HER2 (Figure 3). The patient with stage IVB dedifferentiated endometrial cancer (P7) exhibited marked preoperative expression of PanCK and GATA3. Re-expression of both molecular markers was observed at the third follow-up (Figure 2), which corresponded to disease progression. The patient was treated initially with adjuvant chemotherapy. After treatment was shifted to immunotherapy with Pembrolizumab and Lenvatinib, dramatic clinical improvement was noted, with a small abdominal residual tumor. Expression of CD13 was also noted at the first follow-up.

P13, diagnosed with stage IIIC1 carcinosarcoma, exhibited initial marked expression of CD13 that persisted throughout the first, second, and third follow-ups. Expression of GATA3/PanCK (Figure 2) was observed at the second and third follow-ups. The patient was later confirmed to have pulmonary metastasis.

P10, diagnosed with stage IIIa endometroid carcinoma, exhibited peritoneal recurrence at the third follow-up, with re-expression of PanCK and GATA3. Initial expression of CD13 was observed. CD13 expression persisted throughout the first and second follow-ups (Figure 2).

P5, diagnosed with stage 1, grade 1 endometroid carcinoma, exhibited preoperative expression of all four molecular markers. At the third follow-up, this patient exhibited re-expression of all markers except HER2, with no clinical evidence of disease at the time of writing. CD13 expression fluctuated; increased expression was observed at the first and fourth follow-ups (Figure 2).

P3, diagnosed with stage IA, grade 1 endometroid carcinoma, exhibited GATA3/HER2/PanCK expression at the fourth follow-up and CD13 expression at the second follow-up (Figure 2). This patient exhibited no clinical evidence of disease at the time of writing.

In all patients that exhibited initial HE4 expression, HE4 expression was undetectable after the first or second follow-up (Figure 2).

The two ovarian cancer patients included P1, diagnosed with stage IA immature teratoma, and P2, diagnosed with stage IA mucinous cytadenocarcinoma. Both patients underwent primary surgical intervention, and both exhibited preoperative HE4 expression (Figure 4). No CTCs were noted at the first follow-up in P2, nor were any CTCs detected at the second follow-up in P1 (Figure 2). Both patients showed no evidence of disease at the time of follow-up, although P1 exhibited slight HE4 expression at the fourth follow-up. Marked CD13 expression was observed in P2 at the first follow-up but was subsequently normalized. In P1, CD13 expression peaked at the second follow-up and gradually returned to normal by the fourth follow-up (Figure 2).

All three breast cancer patients had AJCC stage II breast cancer. P8, who had triple negative breast cancer, exhibited marked expression of HER2 and PanCK. P15 and P16 exhibited moderate expression of HER2 (Figure 5). GATA3 re-expression was observed in P8 at the second follow-up and in P16 at the first follow-up. HER2 and PanCK expression was also noted at the first follow-up in P16 and second follow-up in P8 (Figure 2). No clinical evidence of disease was noted in any of the patients at the time of writing.

The patient with vaginal small cell carcinoma exhibited marked expression of GATA3, HER2, and PanCK at the first follow-up (Figure 6), which corresponded to a poor tumor response to initial chemotherapy.

A genomic mutation in TP53 was observed in P10, P11, and P14. A mutation in CDH1 was observed in P3 and P5. Two breast cancer patients (P8 and P16) had the BRCA1 mutation (Table 6).

## 3. Discussion

All seventeen patients in our study cohort expressed cancer-specific molecular markers in CTCs, which were detected using V-Biochip microfluidic device technology in conjunction with an automated platform. The importance of identifying recurrence even before clinical evidence of disease was clearly demonstrated in both cervical and endometrial cancer in this study. CD1, an epithelial mesenchymal transition marker, along with the epithelial marker EpCAM, helped to facilitate the detection, monitoring, and prognostication of gynecologic cancer.

The prognostic role of CTCs in two of the cervical cancer patients was clearly demonstrated in this small series study. These patients exhibited marked expression of HER2 at the fourth follow-up, which led to the confirmation of a metastatic lesion in P4 (endocervical adenocarcinoma). P18 (squamous cell carcinoma) demonstrated persistent expression of CD13 and mild expression of GATA3/PanCK at the second follow-up, which corresponded to avid uptake in the neck lymph node and histological confirmation of metastasis. P19 exhibited marked expression of GATA3/PanCK/HER2 and re-expression of CD13 at follow-up but demonstrated no evidence of disease. Cancer type-specific molecular expression was observed in these three cases, with marked expression of HER2 in endocervical adenocarcinoma and GATA3 in squamous cell carcinoma. All of the confirmed recurrent cases exhibited elevated expression of CD13 at follow-up. Therefore, monitoring CD13 expression in addition to the expression of tumor-specific molecules may be beneficial [10,11,13].

The eight endometrial cancer cases included one carcinosarcoma case and one dedifferentiated case. Both of these cases exhibited marked expression of GATA3/PanCk and CD13 at follow-up, and recurrent disease was confirmed. The re-expression of markers in P3 and P5 warranted further evaluation. The presence of the genomic mutation in CDH1 indicates that the evaluation of hereditary diffuse gastric cancer and lobular breast cancer should be considered when managing follow-up [7,8,9,10,12].

Single-cell genomics and transcriptomics are fields in which further research is necessary [3,4,5,6,7]. The TP53 mutation, observed in P10, P11, and P14, corresponded to metastasis at follow-up. As these three endometrial cancer patients exhibited a poor prognosis, the presence of TP53 mutations should be considered when making disease management decisions.

Cells with PanCK expression were detected in some donors in the healthy group. It is possible that normal endometrial cells with PanCK expression [16] entered the circulatory system; these cells are known as circulating endometrial cells [17,18]. Therefore, in this study, PanCK was deemed unsuitable for use as a target marker.

Microfluidic technology has the advantages of high capture rate and easy operation [19,20,21,22], especially the Cell RevealTM platform with V-BioChip used in this study. In addition to the high capture rate, it can locate the cell position with AI image recognition software, which can pick up CTCs more accurately and provide high-purity target cell biological information for subsequent molecular analysis.The V-Biochip-based CTC detection technique developed in our study increases the contact area with target cells. The high detection rate of CTCs in all patients verified the high sensitivity of this method. Downstream single-cell genomic profiling and detection of the CD13 marker facilitated further in-depth analysis of CTCs.

The limitations of this study included the small number of patients and the short follow-up period, making further investiagion into the generalizability of its utility in gynecologic cancer necessary. Studies involving larger cohorts that include patients with additional heterogeneous diseases are needed.

In conclusion, our novel CTC detection system, based on microfluidic device technology and an automated platform, enabled detection of the expression of specific epithelial and mesenchymal markers in CTCs. This technology can be used for prognostication and early detection of disease recurrence.

## 4. Materials and Methods

### 4.1. Patient Characteristics

From 21 April 2021 to 23 May 2022, seventeen patients (four cervical cancer, six endometrial cancer, three ovarian cancer, one vaginal cancer, and three breast cancer) were enrolled in our study. The first follow-up was arranged at 3 months post-operation or 15 weeks after CCRT/RT/CT. The second, third, and fourth follow-ups were conducted at 3-month intervals. (Figure 7). IRB approval (#110016) was obtained from our hospital before the study. The tumor markers of all patients were analyzed according to their specific cancer histology pre and post treatment. Annual CT scans were arranged, and tumor marker analysis, physical examinations, and optional ultrasoundere performed every 3 months during the first year of follow-up.

### 4.2. CTC Detection Platform and Workflow

Instead of the traditional magnetic bead system used for the retrieval of CTCs, this study employed the Cell Reveal^TM^ (CytoAurora Biotechnologies, Inc., Hsinchu, Taiwan) platform, which is a fully automated platform that uses the V-BioChip microfluidic device for the enrichment and staining of circulating rare cells. The core technology, V-BioChip, is a silicon-based chip produced by metal-assisted chemical etching. The protruding nanostructures of the V-BioChip device are regularly arranged and function in cooperation with the interspace region. The chip undergoes surface treatment, which includes silane-PEG-Biotin deposition and covalent conjugation of streptavidin. This design increases the contact area with target cells. As a result, the target cells can attach to the outer portions of the protruding nanostructures without sustaining punctures or scratches.

The workflow of CTC detection is divided into several steps: collection and preprocessing of the blood sample, enrichment and immunofluorescence staining of CTCs, and scanning and identification of CTCs. Subsequently, target CTCs are isolated with high purity using an automatic cell picker for single-cell whole genome amplification. Figure 8 demonstrates the laboratory workflow of the study.

### 4.3. Sample Collection

For sample collection, 18 mL of peripheral blood was obtained from the patients. The first 2 mL was collected in BD vacutainer K2 EDTA tubes, and the remaining 16 mL was collected in two BD vacutainer ACD tubes (8 mL per tube). Whole blood samples were purified using Lymphoprep^TM^ density gradient medium (STEMCELL Technologies, Vancouver, BC, Canada) for the enrichment of the peripheral blood mononuclear cell (PBMC) fraction.

Isolated PBMCs were fixed with 4% paraformaldehyde for 15 min at room temperature. Fixed PBMCs were then treated with an antibody cocktail containing biotinylated anti-EpCAM antibody (R&D Systems, Minneapolis, MN, USA) and biotinylated anti-E-cadherin antibody (R&D Systems, Minneapolis, MN, USA) and mixed consistently for 30 min at 37 °C. Then, 3 mL of Dulbecco’s phosphate-buffered saline was added to the mixture, which was centrifuged at 400× *g* for 5 min to collect the cell pellets and remove the supernatant.

### 4.4. CTC Enrichment and Identification

Cell Reveal^TM^ (CytoAurora Biotechnologies, Inc., Hsinchu, Taiwan) was used for the enrichment and staining of CTCs. After placing the required reagents in the machine and setting the experimental conditions, the prepared blood sample was injected into the instrument, and the entire process proceeded automatically. The input blood sample was fixed in 4% paraformaldehyde and mixed with 0.1% Triton X-100 (ThermoFisher, Waltham, MA, USA) and 2% bovine serum albumin to increase the cellular permeability. Subsequently, the sample passed through the V-BioChip device at a flow rate of 0.6 mL/h, allowing the target cells to be captured by the chip.

The CTC target cells were identified using four different antibody cocktails, including CD13/EPCAM, HE4/EPCAM, Her2/Gata3/PanCK, and PAX8/EPCAM. Nuclei were stained with 4,6-diamidino-2-phenylindole (Invitrogen, Carlsbad, CA, USA).

After the completion of cell staining, the V-BioChip was moved to a fluorescence microscope, which was controlled by an automated scanning system (CytoAcq Images system, CytoAurora Biotechnologies, Inc., Hsinchu, Taiwan) for whole chip image acquisition. The Cell Analysis Tools (CAT; CytoAurora Biotechnologies, Inc., Hsinchu, Taiwan) system is a tool for cell identification based on image recognition of immunofluorescence staining. The CAT system can screen an entire image within 10 min, identify the target cells, and record the exact position of the target cells on the chip.

### 4.5. CTC Isolation

The target cells were isolated using Cell Picker (CytoAurora Biotechnologies, Inc., Hsinchu, Taiwan) and then dispensed into a PCR tube for whole genome amplification. Cell Picker is a system that integrates a motorized upright fluorescence microscope and a micropipette module (Figure 9). The Cell Picker system can pick single target cells accurately and rapidly according to the target cell location information, which is recorded by the CAT system. The glass needle picks the target cells and deposits them into an Eppendorf PCR tube containing 4 uL of Tris-EDTA buffer.

### 4.6. Whole Genome Amplification

DNA from the CTCs was amplified using the PicoPLEX Single-Cell WGA Kit (Takara Bio, Mountain View, CA, USA). During each whole genome amplification assay, positive control DNA and a no-template control were used to monitor the amplification efficiency and contamination. After whole genome amplification, the DNA was purified using the QIAPrep^®^ Spin Miniprep kit (Qiagen Inc., Valencia, CA, USA). The concentration and purity of the purified DNA were determined using Nanodrop 2000 (ThermoFisher, Waltham, MA, USA), and the size distribution was measured using the Agilent 4200 TapeStation with the Genomic DNA ScreenTape assay (Agilent Technologies, Santa Clara, CA, USA).

### 4.7. PCR-Based Targeted Sequencing

Targeted sequencing was performed using the SureSelect Cancer All-In-One Solid Tumor Assay (Agilent Technologies, Santa Clara, CA, USA), which includes the following 98 cancer genes: ABL1, AKT1, ALK, APC, AR, ARAF, ARID1A, ATM, BCL2, BCR, BRAF, BRCA1, BRCA2, CCND1, CCND2, CCNE1, CD274, CDH1, CDK4, CDK6, CDKN2A, CDKN2B, CIC, CSF1R, CTNNB1, DDR2, DNMT3A, EGFR, ERBB2, ERBB3, ERBB4, ESR1, ETV1, ETV4, ETV6, EZH2, FBXW7, FGR1, FGR2, FGR3, FGR4, FOXL2, GNA11, GNAQ, GNAS, HNF1A, HRAS, IDH1, IDH2, JAK2, JAK3, KDR, KIT, KMT2A, KRAS, MAP2K1, MAP2K2, MAP2K4, MDM2, MET, MLH1, MSH2, MSH6, MTOR, MYC, MYCN, MYD88, NF1, NF2, NFE2L2, NOTCH1, NRAS, NTRK1, PDGFRA, PDGFRB, PIK3CA, PIK3R1, PTCH1, PTEN, PTPN11, RAF1, RB1, RET, RIT1, ROS1, SMD4, SMARCB1, SMO, SRC, STK11, TERT, TMPRSS2, TP53, TSC1, TSC2, VEGFA, VHL, and WT1.

### 4.8. Next-Generation Sequencing Analysis

The target CTCs were isolated using the Cell Picker (CytoAurora Biotechnologies, Inc., Hsinchu, Taiwan) and then dispensed into a PCR tube containing Tris-EDTA buffer for whole genome amplification. After whole genome amplification, the DNA sample was analyzed using the SureSelect Cancer All-In-One Solid Tumor (16 rxn, index 1–16; Agilent Technologies, #G9704S). The enriched DNA was subjected to next-generation sequencing using Illumina NovaSeq 6000 in a 2 × 150 bp format. The average coverage depth of the captured region was 1000× for CTCs and 50× for germline controls.

### 4.9. Statistical Analysis

Descriptive statistics were used to analyze the demographic variables. Continuous variables were expressed as the mean and range, and categorical variables were expressed numerically as percentages. A *p*-value of <0.05 was considered statistically significant. All statistical analyses were performed using R (R Core Team (2019). R: A language and environment for statistical computing. R Foundation for Statistical Computing, Vienna, Austria. https://www.R-project.org/ (accessed on 5 July 2019)).

## Figures and Tables

**Figure 1 ijms-24-02300-f001:**
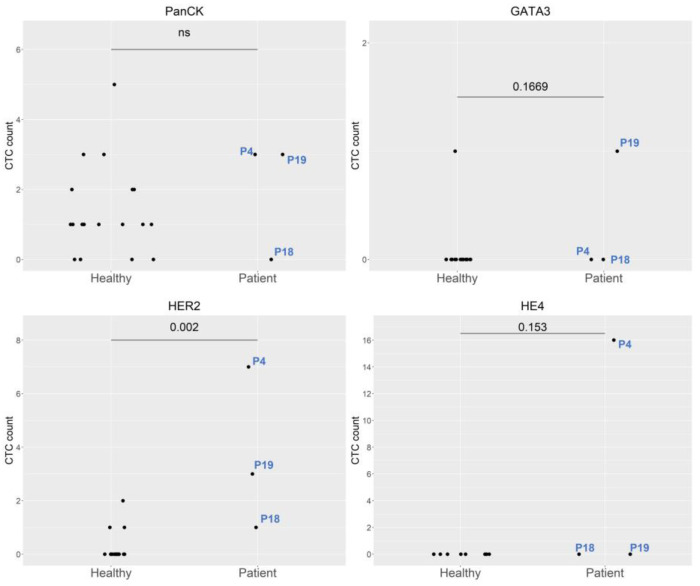
Comparison of molecular expression between healthy controls and cervical cancer patients. Abbreviations: P4, Patient 4; P18, Patient 18; and P19, Patient 19.

**Figure 2 ijms-24-02300-f002:**
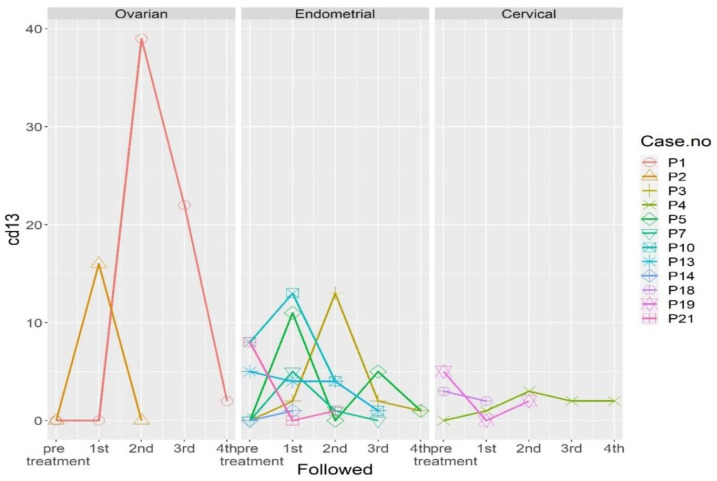
Dynamic changes in the expression of different molecular markers in the 17 patients before and after treatment. Abbreviations: P1, Patient 1; P2, Patient 2; P3, Patient 3; P4, Patient4; P5, Patient 5; P7, Patient 7; P8, Patient 8; P10, Patient 10; P13, Patient 13; P14, Patient 14; P16, Patient 16; P18, Patient 18; P19, Patient 19; and P21, Patient 21.

**Figure 3 ijms-24-02300-f003:**
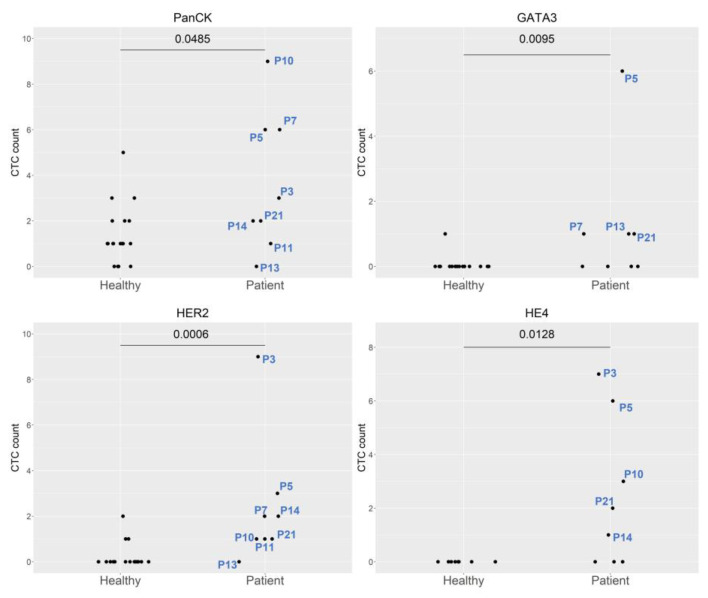
Comparison of the expression of different molecular markers between healthy controls and endometrial cancer patients. Abbreviations: P3, Patient 3; P5, Patient 5; P7, Patient 7; P10, Patient 10; P11, Patient 11; P13, Patient 13; P14, Patient 14; and P21, Patient 21.

**Figure 4 ijms-24-02300-f004:**
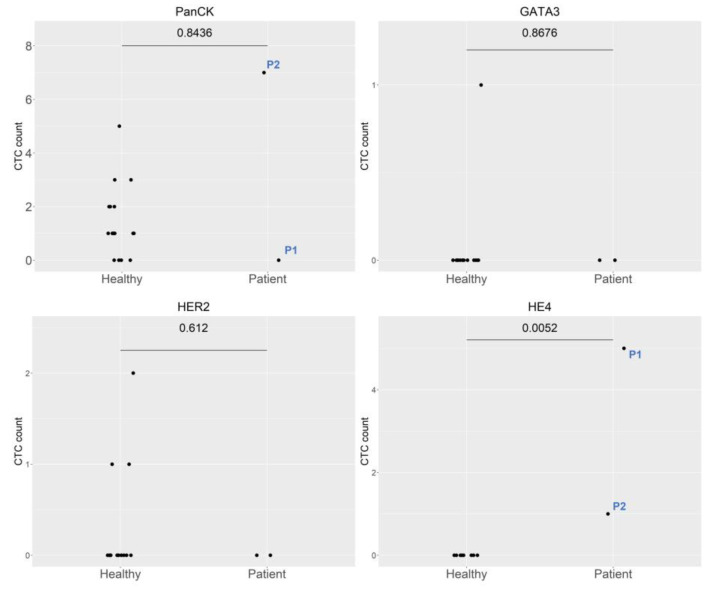
Comparison of the expression of different molecular markers between healthy controls and ovarian cancer patients. Abbreviations: P1, Patient 1 and P2, Patient 2.

**Figure 5 ijms-24-02300-f005:**
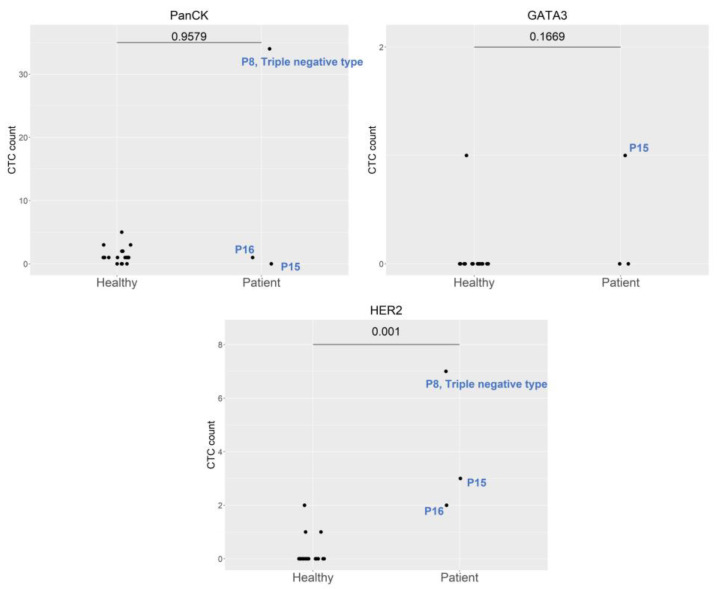
Comparison of the expression of different molecular markers between healthy controls and breast cancer patients. Abbreviations: P8, Patient 8; P15, Patient 15; and P16, Patient 16.

**Figure 6 ijms-24-02300-f006:**
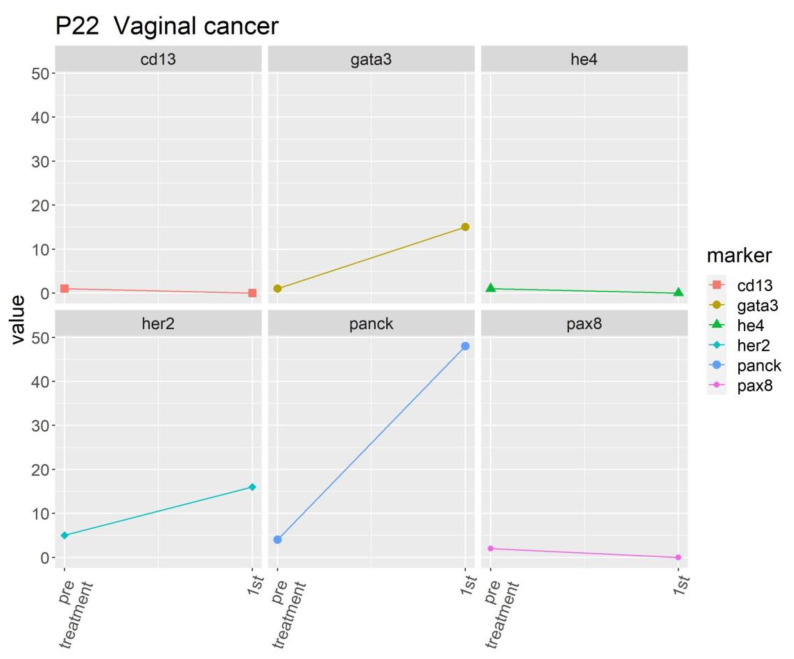
Expression of different molecular markers in a patient with vaginal small cell carcinoma.

**Figure 7 ijms-24-02300-f007:**
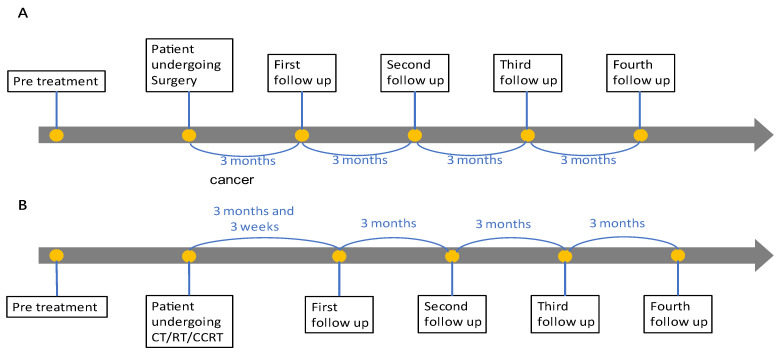
Different follow-up schemes for (**A**) surgical intervention and (**B**) other treatment modalities.

**Figure 8 ijms-24-02300-f008:**
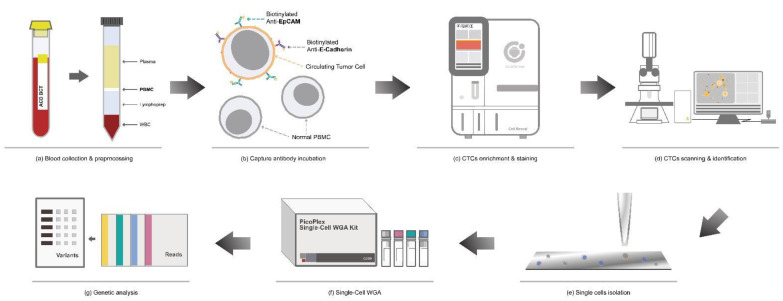
Schematic workflow of circulating tumor cell (CTC) enrichment and characterization. (**a**) Density gradient centrifugation was used to isolate peripheral blood mononuclear cells (PBMCs) from the blood sample. (**b**) PBMCs were incubated with biotinylated antibodies. (**c**) CTCs were enriched and stained via Cell Reveal^TM^. (**d**) The whole chip image was acquired via an automatic scanning system controlled by CytoAcqImages software. Cell Analysis Tools was used to identify the target cells, record their position, and document their morphology. (**e**–**g**) Using Cell Picker, high-purity single cells were isolated for whole genome amplification (WGA) and genomic analysis.

**Figure 9 ijms-24-02300-f009:**
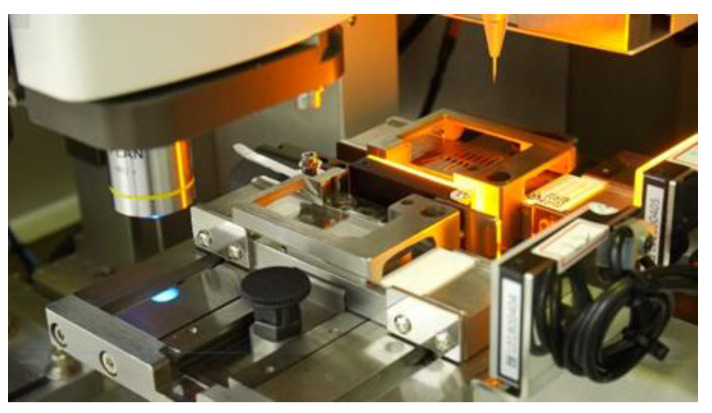
Cell picker.

**Table 1 ijms-24-02300-t001:** Patient demographic characteristics. Patients were numbered according to the date of enrollment.

Case No.	Age	Height	Weight	Marital Status	G	P	A	Admission	Discharge	Hospitalization	Treatment	Treatment Date	Blood Loss	Adjuvant Treatment	Cancer	AJCC Stage	FIGO Stage	Histological Type	AJCC Histological Grade	Retrieved Pelvic Lymph Nodes	Tumor Size (cm)
P1	35	162	49	Single	0	0	0	20210420	20210424	2	Staging laparotomy	20210422	200		Ovarian	1A	1A	Immature teratoma	2	9	15 × 11 × 6.8
P2	39	163	61.5	Married	4	2	2	20210522	20210527	3	Staging laparotomy	20210524	100		Ovarian	1A	1A	Mucinous adenocarcinoma	1	16	21.3 × 19 × 9.5
P3	50	159	57.7	Married	3	2	1	20210714	20210720	5	da Vinci staging	20210715	300		Endometrial	1A	1A	Endometrioid carcinoma	1		10 × 7 × 4.2
P4	52	146	44.5	Married		3		20210728	20210807	9	surgery	20210729	600	CCRT	Cervical	3C1	3C1	Endocervical adenocarcinoma	2	10	5.3 × 3.5 × 2
P5	70	159	61	Married	3	2	1	20210803	20210809	4	laparoscopic staging	20210805	50		Endometrial	1A	1A	Endometroid adenocarcinoma	1	32	3 × 1.7
P6	38														Non cancer						
P7	56	160	52	Single		0		20210822	20210904	12	Staging laparotomy	20210823	2900	CT	Endometrial	4B	4B	Dedifferentiated carcinoma	2		
P8	32	167.6	68.1	Married	1	1	0				CT	20210928		Surgery + RT	Breast	2		Invasive breast carcinoma	3		
P9	79														Non cancer			Squamous metaplasia and endocervical polyp			
P10	67	158	45	Married	3	3	0	20211003	20211013	8	Staging laparotomy	20211005	3200	CT	Endometrial	3A	3A	Clear cell adenocarcinoma	3	38	8.5 × 5 × 4
P11	44	165	68	Married	2	1	1	20211016	20211102	15	Staging laparotomy	20211018	300		Endometrial	3C2	3C2	Endometroid adenocarcinoma	2	9	8 × 5
P12	56														Drop Out						
P13	62	160.5	57.1	Married	3	3	0	20211112	20211123	8	Surgery	20211115	400	CCRT	Endometrial	1B	1B	Carcinosarcoma	3	10	5 × 4
P14	51	146	41.9	Married	5	3	2	20211116	20211130		da Vinci staging	20211118	150	RT	Endometrial	1B	1B	Endometrioid carcinoma	2	11	3 × 1.5
P15	56	170	65	Married		2		20220106	20220112	5	Surgery	20220107	150	CT	Breast	2A	Tis	Invasive breast carcinoma	1	13	2 × 0.9
P16	41	162	90	Married		1					CT	20211229		Surgery + RT	Breast	2B	2B	Invasive breast carcinoma	2	12	1 × 0.6
P17															Drop Out						
P18	55	147	48	Married		4					CCRT	20220214		Surgery	Cervical	3C1	3C1	Squamous cell carcinoma	3		
P19	65	153	81.2	Married		3					CCRT	20220215			Cervical	2B	2B	Squamous cell carcinoma	2		
P20															Drop Out						
P21	57	157	61.5	Married	6	4	2	20220429	20220503	3	Laparoscopic staging	20220430	50		Endometrial	1A	1B	Endometrioid carcinoma	2	29	1.8 × 1.3
P22	75	150.4	65.3	Married		4					CCRT	20220525			Vaginal		3	Small cell neuroendocrine carcinoma			
P23	29	168	52	Married	7	4	3				CT	20220330		Surgery + RT	Hemolysis	4B	4B	Squamous cell carcinoma			

**Table 2 ijms-24-02300-t002:** Demographics of the study population.

Parameter	Cervical *n* = 3	Endometrial *n* = 8	Ovarian *n* = 2	Breast *n* = 3	Vaginal *n* = 1
Age, years	57.33 (52–65)	57.12 (44–70)	37 (35–39)	43 (32–56)	75 (75–75)
Height, cm	148.67 (146–153)	158.06 (146–165)	162.5 (162–163)	166.53 (162–170)	150.4 (150.4–150.4)
Weight, kg	57.9 (44.5–81.2)	55.52 (41.9–68)	55.25 (49−61.5)	74.37 (65−90)	65.3 (65.3−65.3)
BMI, kg/m^2^	25.93 (20.88−34.69)	22.13 (18.03−24.98)	20.91 (18.67−23.15)	27.01 (22.49−34.29)	28.87 (28.87−28.87)
Married	3 (100)	7 (87.5)	1 (50)	3 (100)	1 (100)
Parity					
Multiparous	3 (100)	7 (87.5)	1 (50)	3 (100)	1 (100)
Nulliparous	0 (0)	1 (12.5)	1 (50)	0 (0)	0 (0)
Treatment					
CCRT	2 (66.67)	0 (0)	0 (0)	0 (0)	1 (100)
CT	0 (0)	0 (0)	0 (0)	2 (66.67)	0 (0)
Surgery	1 (33.33)	8 (100)	2 (100)	1 (33.33)	0 (0)
Surgery					
Laterality					
BSO	1 (100)	8 (100)	0 (0)	0 (0)	N/A
RSO	0 (0)	0 (0)	2 (100)	0 (0)	N/A
Rt. Breast	0 (0)	0 (0)	0 (0)	1 (100)	N/A
Hospital day	9 (9–9)	7.86 (3–15)	2.5 (2–3)	5 (5–5)	N/A
Blood loss	600 (600–600)	918.75 (50–3200)	150 (100–200)	150 (150–150)	N/A
Adjuvant treatment					
CCRT	1 (50)	1 (25)	Na	0 (0)	N/A
CT	0 (0)	2 (50)	Na	1 (33.33)	N/A
RT	0 (0)	1 (25)	Na	0 (0)	N/A
Surgery	1 (50)	0 (0)	Na	0 (0)	N/A
Surgery & RT	0 (0)	0 (0)	Na	2 (66.67)	N/A

**Table 3 ijms-24-02300-t003:** Lymph node status and grading of the study population.

Parameter	Cervical *n* = 3	Endometrial *n* = 8	Ovarian *n* = 2	Breast *n* = 3	Vaginal *n* = 1
Lymph node	10 (10–10)	21.5 (9–38)	12.5 (9–16)	12.5 (12–13)	N/A
Stage					
I	0 (0)	5 (62.5)	2 (100)	0 (0)	0 (0)
II	1 (33.33)	0 (0)	0 (0)	3 (100)	0 (0)
III	2 (66.67)	2 (25)	0 (0)	0 (0)	1 (100)
IV	0 (0)	1 (12.5)	0 (0)	0 (0)	0 (0)
Grade					
1	0 (0)	2 (25)	1 (50)	1 (33.33)	N/A
2	2 (66.67)	4 (50)	1 (50)	1 (33.33)	N/A
3	1 (33.33)	2 (25)	0 (0)	1 (33.33)	N/A

**Table 4 ijms-24-02300-t004:** Histological types of the study population.

Breast Cancer	
Invasive breast carcinoma, NST	1 (33.33)
Ductal carcinoma in situ, cribriform type	1 (33.33)
Invasive ductal carcinoma mixed invasive lobular carcinoma	1 (33.33)
**Cervical cancer**	
Endocervical adenocarcinoma	1 (33.33)
Squamous cell carcinoma	2 (67.67)
**Endometrial cancer**	
Carcinosarcoma	1 (12.5)
Clear cell adenocarcinoma	1 (12.5)
Dedifferentiated carcinoma	1 (12.5)
Endometrioid carcinoma/ Endometroid adenocarcinoma	5 (62.5)
**Ovarian cancer**	
Immature teratoma	1 (50)
Mucinous adenocarcinoma	1 (50)
Vaginal cancer	
Small cell neuroendocrine carcinoma	1 (100)

**Table 5 ijms-24-02300-t005:** Circulating tumor cell counts of the study population.

Parameter	Cervical *n* = 3	Endometrial *n* = 8	Ovarian *n* = 2	Breast *n* = 3	Vaginal *n* = 1
CD13	2.67 (0–5)	3 (0–8)	0 (0–0)	N/A	1 (1–1)
HE4	5.33 (0–16)	2.38 (0–7)	3 (1–5)	N/A	1 (1–1)
HER2	3.67 (1–7)	2.38 (0–9)	0 (0–0)	4 (2–7)	5 (5–5)
GATA3	0.33 (0–1)	1.12 (0–6)	0 (0–0)	0.33 (0–1)	1 (1–1)
PanCK	2 (0–3)	3.62 (0–9)	3.5 (0–7)	11.67 (0–34)	4 (4–4)
Pax8	0 (0–0)	0.12 (0–1)	0 (0–0)	N/A	2 (2–2)

**Table 6 ijms-24-02300-t006:** Genomic expression of seven patients.

Cancer	Patient	Gene	Variation	VAF
Endometrial Endometrioid Carcinoma	P3	CDH1	p.R74 *	0.002
		TP53	p.C238R	0.008
		PIK3CA	p.G106R	0.015
		PIK3CA	p.H1047L	0.004
		ESR1	p.Q375H	0.0002
	P5	CDH1	p.R74 *	0.001
		AR	p.M788V	0.005
	P11	ERBB3	p.D297V	0.004
		TP53	p.Y205C	0.018
		AR	p.N706S	0.008
	P14	TP53	p.Y205C	0.011
		CTNNB1	p.S45P	0.008
	P10	NRAS	p.G12D	0.014
		TP53	p.Y205C	0.039
		AR	p.L617P	0.014
		AR	p.A871V	0.003
		AR	p.V890M	0.007
Endocervical adenocarcinoma	P4	FGFR2	p.I548V	0.05
		PIK3CA	p.G118D	0.02
		AR	p.K633 *	0.008
	P18	FGFR2	p.E566G	0.015
		FGFR2	p.K310R	0.007
Breast cancer	P8	ATM	p.R248 *	0.008
		ERBB3	p.D297V	0.003
		BRCA2	p.P704fs	0.004
		BRCA2	p.G1006 *	0.08
		BRCA2	p.L1390fs	0.04
		BRCA2	p.K1691fs	0.025
		BRCA2	p.1862ins	0.013
		BRCA2	p.E2020 *	0.009
		BRCA2	p.F2254fs	0.075
		BRCA1	p.R1772 *	0.003
		BRCA1	p.K1771fs	0.002
		BRCA1	p.G1759R	0.02
		BRCA1	p.Q1313 *	0.01
		BRCA1	p.K1110fs	0.017
		BRCA1	p.Q934 *	0.007
		BRCA1	p.Q759 *	0.04
		BRCA1	p.K654fs	0.05
		BRCA1	p.K614 *	0.005
		BRCA1	p.W385 *	0.008
		BRCA1	p.K339fs	0.024
		BRCA1	p.E149 *	0.003
	P16	BRCA2	p.Q407 * fs	0.007
		BRCA2	p.D559 * fs	0.013
		BRCA2	p.S1442 *	0.156
		BRCA1	p.K1814 *	0.01
		BRCA1	p.G1759 *	0.025
		BRCA1	p.K1711 *	0.031
		BRCA1	p.E1556 *	0.017
		BRCA1	p.I917fs	0.007
		BRCA1	p.K654fs	0.038
		BRCA1	p. L30 *	0.02

* means translation termination (stop) codon.

## Data Availability

Data is unavailable due to privacy.

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
