# Peer review of "Detection of Circulating Tumor Cell-Related Markers in Gynecologic Cancer Using Microfluidic Devices: A Pilot Study"

_ijms, 2023, doi:10.3390/ijms24032300_

Round 1

Reviewer 1 Report

The Law and coll work, despite the fact that it has a potential scientific interest, since the characterization and clinical validation of the study of CTCs in tumors such as endometrial and ovarian cancer remains a challenge, in the current state it does not have enough impact to be published. The results that are presented are not very cohesive and integrated, the graphs are difficult to follow and the results are hardly discussed, at least in a way that is not merely descriptive. I suggest the authors make an effort to better define and describe the objectives, results and conclusions of the pilot study, which could be of interest to the reader with a good presentation. There are many aspects regarding the results obtained not mentioned at all, such as the positivity found for several markers in healthy donors. This is just and example, but in general, I recommend and important work to improve the quality of the manuscript.

Reviewer 2 Report

Gynecological cancers have been a burden and their early detection and proper treatment are crucial. The current study is tackling an interesting topic for the readers. Biomarker-driven diagnostic and prognostic approaches are critical for confronting cancers. The manuscript is well-written, and the language is clear. Some things that could be considered:

1-     Did you manage to get samples from different races of patients, or all are from the same race? Could be a useful item to add to the demographics of patients

2- In line 131 it is stated under (PCR-Based Targeted Sequencing): ‘which includes the following 90 cancer genes’, it is better to include the table number for clarity

3- Minor grammatical, punctuation, and typing errors like for instance line 35: Circulating Tumor Cell (CTSs) should be (CTCs), line 44: long space

Round 2

Reviewer 1 Report

Although the authors made an effort to improve the manuscript, the scientific impact and the quality of the work are not enough to be pusblished.

Author Response

Thanks for your comment. We have make changes in our Title, abstract as well as adding a few sentences on the limitation paragraph under the discussion section.

Reviewer 2 Report

Accept

Author Response

Thanks again for your kindly recommendation and acceptance of this article.
